# The Incorporation of Graphene Nanoplatelets in Tung Oil–Urea Formaldehyde Microcapsules: A Paradigm Shift in Physicochemical Enhancement

**DOI:** 10.3390/polym16070909

**Published:** 2024-03-26

**Authors:** Abdullah Naseer Mustapha, Maitha AlMheiri, Nujood AlShehhi, Nitul Rajput, Zineb Matouk, Nataša Tomić

**Affiliations:** Advanced Materials Research Centre (AMRC), Technology Innovation Institute (TII), Masdar City, Abu Dhabi P.O. Box 9639, United Arab Emirates; maitha.almheiri@tii.ae (M.A.); nujood.alshehhi@tii.ae (N.A.); nitul.rajput@tii.ae (N.R.); zineb.matouk@tii.ae (Z.M.); natasa.tomic@tii.ae (N.T.)

**Keywords:** microcapsules, synthesis, in situ polymerization, graphene nanoplatelets, tung oil, self-healing

## Abstract

Tung oil (TO) microcapsules (MCs) with a poly(urea-formaldehyde) (PUF) shell were synthesized via one-step in situ polymerization, with the addition of graphene nanoplatelets (GNPs) (1–5 wt. %). The synergistic effects of emulsifiers between gelatin (gel) and Tween 80 were observed, with gel chosen to formulate the MCs due to its enhanced droplet stability. SEM images then displayed an increased shell roughness of the TO-GNP MCs in comparison to the pure TO MCs due to the GNP species on the shell. At the same time, high-resolution transmission electron microscopy (TEM) images also confirmed the presence of GNPs on the outer layer of the MCs, with the stacked graphene layers composed of 5–7 layers with an interlayer distance of ~0.37 nm. Cross-sectional TEM imaging of the MCs also confirmed the successful encapsulation of the GNPs in the core of the MCs. Micromanipulation measurements displayed that the 5% GNPs increased the toughness by 71% compared to the pure TO MCs, due to the reduction in the fractional free volume of the core material. When the MCs were dispersed in an epoxy coating and applied on a metallic substrate, excellent healing capacities of up to 93% were observed for the 5% GNP samples, and 87% for the pure TO MC coatings. The coatings also exhibited excellent corrosion resistance for all samples up to 7 days, with the GNP samples offering a more strenuous path for the corrosive agents.

## 1. Introduction

Drying oils are a class of materials that can react with atmospheric oxygen to cross-link and solidify. In this case, polymerization can occur without the use of a catalyst, owing to the interaction of oxygen in the air. The unsaturated conjugated systems in these oils’ structures are frequently related to their quick curing (the cross-linking of polymer chains). The degree of unsaturation determines the oil’s curing pace, with higher degrees of unsaturation resulting in quicker cross-linking. TO is a type of drying oil, and it has been used in a variety of applications, including varnishes, paints, printing inks, and oil cloths, due to its propensity to polymerize into a durable solid film [1,2,3].

To circumvent the drying oil from cross-linking prematurely and to create a barrier with oxygen, they can be microencapsulated in a suitable shell material. In situ polymerization is a well-established chemical microencapsulation method in which a polycondensation reaction occurs to create a shell material around an active core material, triggered by a change in pH and temperature [4,5]. Amino resins are popularly utilized as shell materials for this process, which include Poly(urea-formaldehyde) (PUF). UF possesses the merits of being chemically stable, having great mechanical stability, water resistance, low permeability, and high core material loading [6].

Various researchers have utilized various additives in MCs, and graphene-based additives offer several advantages [7,8,9]. Recent studies have revealed that the incorporation of graphene improves the corrosion resistance of the polymer coatings. The enhancement of the coating performance is due to the excellent properties of graphene, such as chemical inertness, impermeability towards corrosive agents such as oxygen, corrosive ions, and water, and hence, the presence of graphene offers a tedious path to the corrosive agents [10,11]. For example, graphene-oxide is a widely used additive in MCs to increase mechanical strength, thermal conductivity, and electrical conductivity. GO was utilized as a stabilizer in a Pickering emulsion to encapsulate a drying oil (linseed oil), in which the chemical stability was increased, as observed when the MCs were dispersed in hexane (linseed oil extraction agent) [7]. Carbon nanotubes have also been used in corrosion mitigation applications and can also be used as potential additives in MCs [12]. However, a less utilized graphene-based additive is graphene nanoplatelets (GNPs), and this material has not been incorporated in MCs containing drying oils before. GNPs are two-dimensional carbon structure materials with single or multiple layers of graphite plane. These materials possess a high electrical conductivity, high modulus, high strength, high thermal conductivity, and a high specific surface area, which are all desirable properties [13]. GNPs are not only a remarkable reinforcing material owing to their morphological structure, which is like monolayer graphene, but they are also low-cost, which widens their application possibilities [14]. For example, they have also been used in microencapsulated phase change materials (MPCMs), resulting in enhanced properties such as a higher thermal conductivity and reduced supercooling [15,16].

In this paper, we propose to incorporate GNPs into the TO MCs, to observe the morphological properties of the MCs, as well as the size distribution and the surface properties, and the effects on the formulation process. Furthermore, the curing behavior of TO with the addition of GNPs is to be explored, as well as the intercalation phenomena. Moreover, the dispersion of the MCs in an epoxy coating to study the self-healing and corrosion resistance are to be investigated.

## 2. Materials and Methods

### 2.1. Materials

The following chemicals were acquired from Sigma-Aldrich (Gillingham, UK): formaldehyde solution (104003, ACS reagent, about 37.0% in solution, resorcinol (398047, ACS reagent, 99.0%), Tween 80 (402788, 99.5%), gelatin (gel) (04055, from porcine skin). Tung oil (TO) (100% pure) was bought from Hopes (Fort Mill, SC, USA). Ammonium chloride (99.5%) was acquired from Daejung, Siheung-si, Republic of Korea. GNP aggregates are aggregates of sub-micron platelets with a diameter of less than 2 µm and a thickness of a few nanometers, a bulk density of 0.2 to 0.4 g cm^−3^, an oxygen content of <2 wt. % and carbon content of >98 wt. %, Brunauer–Emmer–Teller (BET) surface area of 750 m^2^ g^−1^, supplied as a black powder from Strem Chemicals, Inc (Newburyport, MA, USA).

### 2.2. MC Formulation

Firstly, the TO and the GNP aggregates were mixed in an ultrasonic processor (Hielscher UP400St, Teltow, Germany) at 250 W and 68 amps, for a duration of 1 min. The microencapsulation of the GNP/TO was carried out via the one-step in situ polymerization process. The emulsifier solutions were pre-prepared before the experiment, by mixing and dissolving 0.5 g of the chosen emulsifier in 150 g of deionized water. Afterwards, using a Kern ABT 100-5NM balance (Chino, CA, USA), 2.5 g urea, 0.25 g resorcinol, and 0.25 g ammonium chloride were measured in the prepared aqueous solution. The solution was then stirred at room temperature, using a Thermo Scientific (Waltham, MA, USA) HPS RT2 Advanced stirrer. This solution was stirred until a completely clear solution was observed. Using a Mettler Toledo (Columbus, OH, USA) SevenCompact Duo pH meter, the pH of the solution was measured to pH 3.5, using diluted 1 mol L^−1^ HCl solution. After this, the solution was placed under a Silverson L5M-A homogenizer at 10,000 rpm, and 10 mL of the TO was injected into the solution. After a 20 min homogenization period, the solution was placed into a LabTech LWB-111D water bath (Sorisole, Italy), at 60 °C, and 6.5 mL of formaldehyde was injected. This temperature was then maintained for 4 h, and after this duration, the final MCs were left to cool down to 25 °C and separated using a separating funnel and a vacuum filter. The filtration process was repeated 4 times with 25 °C deionized water. The MCs were then left to dry overnight in ambient conditions. Figure 1 represents a schematic diagram for this process and the MCs with the incorporated GNPs.

### 2.3. Characterization Methods

#### 2.3.1. Rheological Properties

The viscosity of TO with and without the addition of different ratios of GNPs was measured following methods described in the literature [17]. The temperature dependence of the viscosity of these oils was determined by using a Brookfield RSO oscillatory Rheometer supplied by Venktron (Dubai, United Arab Emirates) in conical plate mode (spindle diameter = 50 mm). The shear rate of 20 s^−1^ was set for the temperature range of 30–100 °C. The duration of each measurement was 10 s. The viscosity was determined at different shear rates, from 10 to 1000 s^−1^ at 35 °C.

#### 2.3.2. Mastersizer

The Malvern Mastersizer 2000 particle analyser (Malvern, UK) with a wet dispersion unit was used to characterize the MC and TO droplet size distributions (Hydro 2000S). As a dispersion, deionized water was employed. Each experiment consisted of quintuplets of samples taken directly from the aqueous solution. The average size distribution was calculated using the Malvern program, which considered the average particle sizes (both surface and volume averages).

#### 2.3.3. Optical Microscopy (OM), Transmission Electron Microscopy (TEM), and Scanning Electron Microscopy (SEM)

An optical microscope (DSX 1000, Münster, Germany) with a DSX10-SXLOB lens was used to obtain bright-field pictures and investigate the morphological properties of the MCs and TO droplets. Further investigation was carried out using electron microscopic techniques. Scanning electron microscope (SEM) imaging of the samples was conducted in a dual beam system, Scios 2 (Thermo Fisher Scientific, Petaluma, CA, USA). The MCs were coated with 5 nm of Chromium using a Quorum Q150T S plus sputter (Laughton, UK) to enhance the conductivity of the samples. Some of the MCs were also dispersed on Cu TEM grids and dried well for the purpose of observing the MCs in TEM. The samples were investigated using an 80–300 kV Cs corrected Titan system.

Furthermore, TEM was also carried out on the cross-sectional samples prepared using the ultramicrotomy method. After air-drying aliquots of the samples for several hours until they appeared to be visibly dry under a dissecting microscope, Spurr’s epoxy resin was added. A low-viscosity epoxy resin embedding medium for electron microscopy was added to the sample vials, and the samples were put into a vacuum oven (10 in·Hg) for an hour to improve infiltration of the resin. Subsequently, the samples were placed into embedding moulds at 60 °C overnight. Thin sections (approx. 80 nm) were cut with a diamond knife on a Leica UCT ultramicrotome and picked up onto Cu grids. The sections were then viewed in a JEOL JEM 1200 EX TEMSCAN transmission electron microscope (Tokyo, Japan) operating at an accelerating voltage of 80 kV. Images were acquired with an AMT 4-megapixel digital camera (Advanced Microscopy Techniques, Woburn, MA, USA).

#### 2.3.4. Raman Spectroscopy

Raman spectroscopy is frequently utilized to provide a structural fingerprint that allows molecules to be recognized, and in this case, it can help determine if the GNP was encapsulated. Raman spectra were recorded using the LabRAM Soleil™ multimodal Raman Microscope (Horiba Scientific, Kyoto, Japan) equipped with two automated built-in lasers (532 nm and 785 nm). An excitation power of 50% (33 mW) was used during the experiments using an optical microscope equipped with a 5× objective. The spectra were acquired in the 300–2700 cm^−1^ range. Each spectrum was obtained by 50 accumulations with 5 s acquisition time each.

#### 2.3.5. Micromanipulation

The mechanical properties of the MCs were measured using a micromanipulation technique based on the parallel plate compression of individual microparticles using a force transducer (Model 403A, Aurora Scientific Inc., Aurora, ON, Canada). A droplet of microparticle suspension was pipetted onto a specially designed highly tempered glass slide, which was left drying in the air for at least 1 h at an ambient temperature of 23.5 ± 1.5 °C before single MCs were tested. A compression speed of 2.0 μm·s^−1^ was selected and used to compress each single MC. The diameter of each microparticle was measured using a side-view camera (AM4023CT, DinoEye C-Mount Camera, Dino-Lite, Hemel Hempstead, UK). Fifty randomly selected individual MCs were compressed to generate statistically representative results

The data of force versus displacement generated from the micromanipulation measurements were analyzed to determine the rupture force (F_r_ and displacement at rupture (δ_r_), which, combined with the diameter (D) were used to calculate the apparent toughness. Moreover, the force versus displacement data up to a fraction deformation (ratio of the displacement to its original diameter) of 10% were used to calculate the apparent Young’s modulus E value of single MC based on the Hertz model and an assumption of incompressible microparticle (Poisson ratio equal to 0.5) and Coefficient of Determination (R2). The detailed data of force versus displacement curves, the fittings using the Hertz model, and the rupture strength parameters are attached.

## 3. Results and Discussion

### 3.1. Rheology

To characterize the rheological behavior of TO and how it changes with the addition of GNP, Figure 2a illustrates the change in the measured viscosity as a function of the shear rate for the TO with different GNP loadings. The Newtonian Equation (1), which is provided as follows, is the fundamental formula used to calculate a fluid’s rheological properties [9]:(1)σ=ηγ
where σ represents shear stress, η represents the viscosity of the fluid, and γ represents the shear rate. A Newtonian fluid will typically keep the viscosity the same at various shear rates or will be shear-rate-independent, according to Equation (1). Based on this equation, it can be observed that the drop in the viscosity with the increasing shear rate indicates the non-Newtonian behavior of TO, which is even more emphasized by the addition of the GNPs. The viscosity dropped by 6% for the pure TO, 7% for the 1% GNP/TO, 8% for the 3% GNP/TO, and 19% for the 5% GNP/TO. It can be noticed that the 1% GNP slightly increases the natural viscosity of the TO, which is even more emphasized with a further increase in the GNP loadings.

The impact of the temperature on the rheological behavior of the TO with and without the GNPs from 40 to 100 °C was also analyzed (Figure 2b). The viscosity curves followed the same trend when the GNPs were added but at higher values of measured viscosity.

### 3.2. Size Distribution of Oil Droplets’ Diameter

To encapsulate the TO-GNP samples, an oil-in-water (O/W) emulsion must first be produced. Conventionally, an O/W emulsion is described as a thermodynamically unstable system made up of two immiscible liquids (often water and oil) in which the oil is dispersed throughout the water [18]. Through creaming, coalescence, flocculation, or Ostwald ripening, emulsions may eventually separate into two phases [19]. To reduce the interfacial tension between the oil and the water phases, emulsifiers/surfactants can be utilized to stabilize the droplets. The two surfactants of choice utilized here include Tween 80 and gelatin (gel). The hydrophilic head of the Tween 80 molecule is directed toward the aqueous phase and the hydrophobic tail toward the oil phase in the Tween 80-stabilized emulsion, generating a film at the oil–water interface during emulsification [20]. Gel is a naturally occurring amphiphilic macromolecule that can serve as an emulsifier in O/W emulsions due to their surface-active characteristics. It has also been seen to reduce the interfacial dilatational viscosity of the interface in O/W emulsions and increase the viscosity of the water phase to reduce the rate of the coalescence of the oil droplets [21,22].

To explore the oil droplets after homogenization in water, the OM images can be seen in Figure 3, in which four emulsions were produced. This includes TO with Tween 80 emulsifier, TO with gel emulsifier, TO and 1 wt. % GNP with Tween 80 emulsifier, and TO and 1 wt. % GNP with gel emulsifier. It can be observed in Figure 3a,b that the Tween 80 and gel produced droplets with distinctively varying droplet sizes, with the gel samples resulting in much smaller droplets. The oil emulsions with the two emulsifiers were then analyzed using a Mastersizer and the results are summarized in Table 1. It can be noticed that gel provided oil droplets of smaller sizes, but Tween 80 contributed to the more uniform size distribution. The addition of the GNPs to the oil at first reduced the size, but with 5 wt. %, the size started increasing again due to the possible aggregation of the GNPs. The reduction in size was attributed to the stabilizing effect of the TO by the addition of the GNPs [23]. Furthermore, the combination of both gel and Tween 80 was investigated for the TO emulsion with 5 wt. % of GNPs. This combination of emulsifiers was observed to result in a further reduction in size, but with reduced uniformity.

### 3.3. The Formulation of TO and GNP/TO MCs

As the gel emulsifier produced samples with smaller droplet sizes, and due to the success of this emulsifier in the literature elsewhere [5,24], the MCs were then formulated with the one-step in situ polymerization technique [6,25]. Interestingly, during the homogenization step for the microencapsulation process, there seemed to be an observable reduction in foaming when the 1–5% GNP was added to the TO. Figure 4 conveys an image during the homogenization process for the pure TO and 1% GNP/TO, showing a clear reduction in the foaming. This benefits the process, as there would be no requirement to add anti-foaming agents, such as octanol, into the aqueous phase. Furthermore, XPS and XRD analysis was carried out to study the effect of TO on the GNP structure (Appendix A).

After the microencapsulation process, as seen in the OM images, all the MCs produced had a similar morphology and size. This can be seen in Figure 5. Additionally, SEM images (Figure 6) were taken to observe the shell roughness, and the inner structure of the MCs. As seen, the pure MCs had a much smoother morphology than the GNP samples, with particles embedded in the shell to give them a rougher appearance. These are assumed to be the GNP particles, which are believed to be embedded in both the shell material and contained within the core.

TEM investigations further provided information related to the morphology of the MC and graphene-nanoparticle-embedded MCs. Figure 7a confirms the smooth spherical shape of the pure MCs. On the other hand, the GNP-incorporated MC (Figure 7b) clearly shows a contrast morphology as compared to the pure MCs, which is in accordance with the SEM observation in Figure 6. Additionally, the high-resolution image in Figure 7c displays the presence of graphene layers on the outer surface of the MCs. The graphene regions observed are composed of 5–7 layers with the interlayer distance of ~0.37 nm (zoomed in images). The fast Fourier transform (FFT) inset in Figure 7c obtained from a reduced area from the MC further indicates the presence of hexagonal patterns corresponding to a graphitic structure.

Furthermore, to investigate whether the GNP species were encapsulated by the MCs, cross-sectional TEM samples were prepared using the ultramicrotomy method from different compositions of the GNP MCs (described in the experimental method section). The obtained TEM images are shown in Figure 8. The MCs can be clearly seen in the images which are of micron sizes. Figure 8a corresponds to MCs without any addition of GNPs. The phase contrast TEM image clearly indicates the absence of any other material other than the tung oil. Furthermore, the average shell thickness for the pure MC is evaluated to be approximately 45 nm. On the other hand, the images Figure 8b–d correspond to the GNP/TO MC compositions of 1% GNP, 3% GNP, and 5% GNP, which clearly indicates the GNP encapsulation in the MC core. Furthermore, the average shell thickness was evaluated to be 71 nm for the 1% GNP/TO MC, 70 nm for the 3% GNP/TO MC, and 67 nm for the 5% GNP/TO MC. It is evident that the addition of the GNP has increased the thickness of the shell by approximately 24 nm compared to the pure MCs. It is to be noted that not all the MCs have a spherical shape; some of them could have deformed during the formation because of the competing growth nature. In certain MCs, the presence of GNPs was also observed to remain outside the MCs, which is in accordance with our earlier observation (Figure 7).

### 3.4. Raman Spectroscopy

To determine the core material content of the microcapsules, Raman spectroscopy was carried out. Initially, the pure TO and pure GNP samples were characterized, as seen in Figure 9. For the pure TO, the peaks at 1238 cm^−1^ represent a C–O bond, while the peaks at 1376 and 1416 cm^−1^ denote C–H stretching vibrations. The peak observed at 1642 cm^−1^ represents C=C bonds, and the broad peak at 2954 cm^−1^ represents C–H bonds. These characteristic peaks are also similar to another study found elsewhere [26]. Regarding the Pure GNP Raman spectra, this conveyed the characteristic D band at 1350 cm^−1^ and G band at 1560 cm^−1^, with the 2D band present at 2690 cm^−1^. The calculated I_D_/I_G_ value of the GNP was 0.73.

Figure 9b conveys the core material of the crushed GNP/TO MCs in order to analyze the core material contents. There are distinctive peaks at 1238, 1350, and 1672 cm^−1^. Upon close observation, there is a minor peak at 1560 cm^−1^, which may be attributed to the G band of the GNP, as well as minor peaks observed at 2690 cm^−1^, which may be attributed to the 2D band. It seems that there may indeed be a certain amount of GNP in the core material, which would supplement the TEM cross-sectional images attained earlier, conveying the core material composition.

### 3.5. Micromanipulation of the MCs

To study the mechanical properties of the MCs, micromanipulation measurements were carried out, in which 50 random MCs per sample were compressed to obtain an accurate representation of the results. To show an example of this compression, Figure 10 conveys the pure MC sample becoming compressed by the probe, which shows the process before and after the compression process. Table 2 then shows the results of the micromanipulation measurements. It is seen that the pure MC exhibited an apparent toughness of 0.52 MPa, and a Young/s modulus of 28 MPa. However, when just 1% GNP was added to the MC, the value of the toughness increased greatly to 0.89 MPa, which was a 71% increase, and the Young’s modulus went to 39 MPa, also increasing compared to the pure sample. The 3% GNP sample had an apparent toughness of 0.82 MPa with a Young’s modulus of 40 MPa, while the 5% GNP sample had a toughness of 0.89 MPa and a Young’s modulus of 38 MPa. Overall, the 1–5% GNP samples all conveyed a large increase in the mechanical strength of the MCs, which would be beneficial for the long-term storage and handling of the MCs, and the long-term containment of the core material. In a similar study [27], the mechanical properties of the PUF MCs incorporated with nano-SiO_2_ were investigated both experimentally and by a molecular dynamics simulation. It was also seen here that the SiO_2_ had a significant effect in augmenting the mechanical properties, similar to the GNPs in this case. In their work, the density of the MC increased, and the fractional free volume reduced, which in turn, greatly enhanced the mechanical properties of the MCs. A similar mechanism is proposed to also happen in this case, leading to the higher apparent toughness of the samples.

### 3.6. Self-Healing and Corrosion Resistance of the Coating

Epoxy resins are widely used in coatings due to their excellent anti-corrosion properties, chemical resistance, and strong paint adhesion. Therefore, the MCs were added into an epoxy resin primer to formulate a self-healing coating. An important consideration with the addition of the MCs are their adhesive properties and how this is affected. Subsequently, pull-off adhesion tests were carried out to see the MCs’ effect on the adhesive properties of the coating on the metallic substrate. Contact angle measurements were also carried out to observe the coating adhesion on the substrate surface (Appendix A). These tests were carried out in triplicate, and the results can be seen in Table 3. As observed with the pure coating without the MCs, the adhesion strength was 12.82 MPa, while the addition of the MCs without graphene (pure) was seen to be 12.61 MPa. With the addition of 1%, 3% and 5% GNP, these results slightly reduced to 12.22, 12.18 and 12.18 MPa, respectively. The addition of the MCs did not reduce the adhesive properties by a significant amount, ultimately alluding that there has not been a clearly discernable negative effect on the primer’s adhesive properties on the substrate.

As the primer with the addition of the MCs seems to display good adhesion to the substrate, scratch tests were then performed to investigate the self-healing properties of the MCs. Figure 11 conveys the initial scratches on the metallic substrates with the coatings applied, and the scratches 24 h after, in which a 3D model was constructed to observe the scratch depth before and after the healing process. Furthermore, Table 4 conveys the healing efficiency of the metallic plates. As observed, the coating with no MCs did not exhibit any clear healing, with a healing efficiency of 3%, while the coating with the TO-only MCs expressed a healing efficiency of 87%. The addition of the GNP MCs exhibited enhanced healing efficiencies, with 91%, 91%, and 93% healing efficiencies for the 1%, 3%, and 5% samples, respectively, resulting in a slight increase compared to the pure TO MC sample. Overall, the addition of the GNP additives increased the healing efficiency of the coating compared to the pure sample, and this coupled with the long-term increase in the mechanical properties of the MCs will prove to be a beneficial synergistic property overall for the coating.

In a similar case, Wu et al. [28] fabricated GO-modified self-healing MCs for cardanol-based epoxy anti-corrosion coatings. In their work, it was proposed that the GO’s distinct hexagonal ring structure gives it an extremely high barrier property, making it a dense 2D material. The microcapsules’ surface will become coated in GO, which will prevent the corrosive medium from passing through them and enhance the barrier properties and enhance the corrosion resistance. In this case, the GNP is also a dense structure that would augment the anti-corrosion properties of the coating, allowing for a more robust barrier to corrosive agents. Therefore, the addition of the GNPs to the MCs to enhance the mechanical properties may make them more resilient to premature damage, which would reduce the likelihood of cracks or ruptures that can happen to potentially compromise the self-healing capability. Gel permeation chromatography (GPC) measurements in the Appendix A also showed an increase in the M_w_ of the MCs with the addition of GNP. 

In terms of the mechanism of how GNPs can affect the self-healing efficiency of the coatings, this can be divided into a few key areas. The excellent physicochemical properties of graphene allow it to contribute to various functional roles during the healing process [29]. For extrinsic self-healing, such as the type carried out in this work, the six-membered carbon structure of graphene in general allows it to attract the repair agents through π-π electron interactions, and the presence of healing agents containing oxygen groups also have adsorption sites for various graphene derivatives, such as GNP [30]. As graphene is an excellent conductor of electricity, due to its unique two-dimensional structure, when the GNPs are added to the coating containing the MCs, a more conductive network within the matrix may then be created compared to the MCs with no GNPs [29,31]. Subsequently, when the coating is scratched, or there is some form of mechanical damage, this more conductive network formed by the GNPs would likely facilitate the movement of electrons through the damaged area. The GNPs can then act as catalysts for the polymerization of the tung oil, which would enhance the speed and efficiency of this process. Therefore, the incorporation of the GNPs not only augments the mechanical strength of the healed region but may also contribute to the chemical aspects of the self-healing process.

The pure coating and the MC-embedded coating were then applied onto the steel substrates. The wt. % (MC content) was 5% of the total composition of the coatings, for all the samples. After the coatings were applied onto the steel substrates, a scratch was penetrated with a scalpel. Subsequently, the samples were immersed in a 3.5 wt. % NaCl solution and kept there for a duration of 1 week. It can be observed in Figure 12 that the sample with the pure coating exhibited significant corrosion due to the lack of protection from the salt solution to the steel substrate. However, the coatings which included the MCs with and without the GNPs conveyed little to no corrosion after 1 week, which was a significant improvement. Generally, the addition of the GNPs did not significantly improve or hinder the corrosion resistance when incorporated into the epoxy coating, with all the samples showing excellent protection. However, the GNPs may elongate the lifespan of the MCs due to the enhanced mechanical strength, which may reduce the likelihood of premature cracking.

## 4. Conclusions

An investigation into the incorporation of GNPs (1–5 wt. %) into TO MCs was carried out in this work. It has been observed that the addition of GNPs resulted in a range of advantageous properties. The emulsifiers gelatin (gel) and Tween 80 had synergistic effects. Gel was used to create the MCs because of their droplet stability. As shown by the high-resolution TEM images, as well as the Raman spectroscopy data, it is evident that the GNPs were present in both the exterior of the shell and inside the core material contents. The GNP MCs were proven to be more robust through micromanipulation measurements, with an increase in the toughness from 0.52 MPa for the pure TO MC, with a sharp rise to 0.89 MPa for as little as 1% GNP, conveying a 71% increase. This was attributed to be because of the reduction in the fractional free volume of the MCs, which in turn augmented their mechanical properties. Furthermore, the self-healing coating demonstrated excellent self-healing efficiencies of 87% for the pure TO MCs, and 91%, 91%, and 93% for the 1%, 3%, and 5% GNP samples, respectively. For all the samples, up to a week, the MCs also demonstrated excellent corrosion resistance.

## Figures and Tables

**Figure 1 polymers-16-00909-f001:**
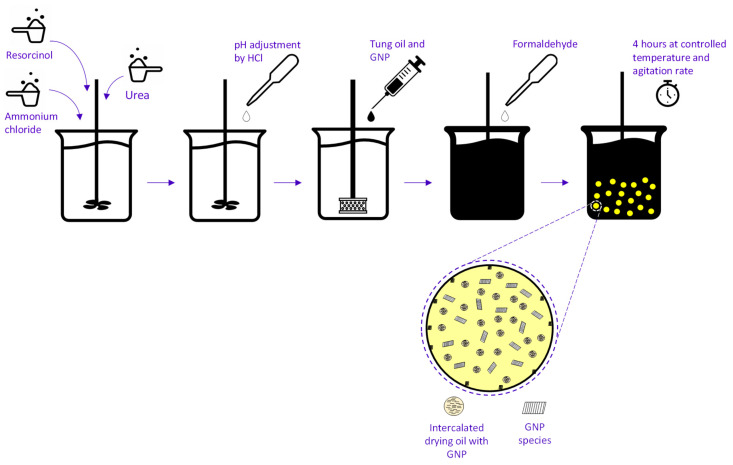
A schematic representation of the formulation process of the MCs and the representation of the MCs with the possible incorporation of GNP.

**Figure 2 polymers-16-00909-f002:**
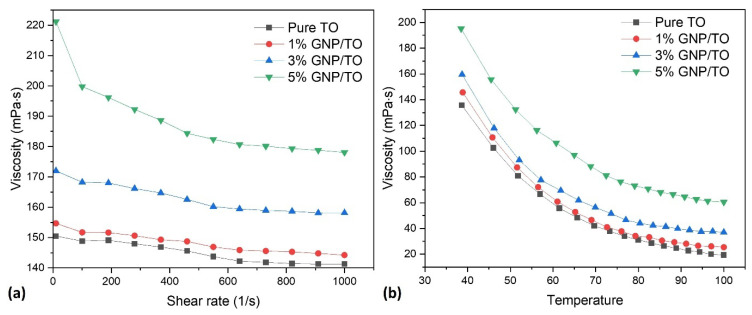
Viscosity profiles of TO with different amounts of GNP: (**a**) with an increasing shear rate (10^−1^ to 10^3^ s^−1^), and (**b**) during heating from 40 to 100 °C.

**Figure 3 polymers-16-00909-f003:**
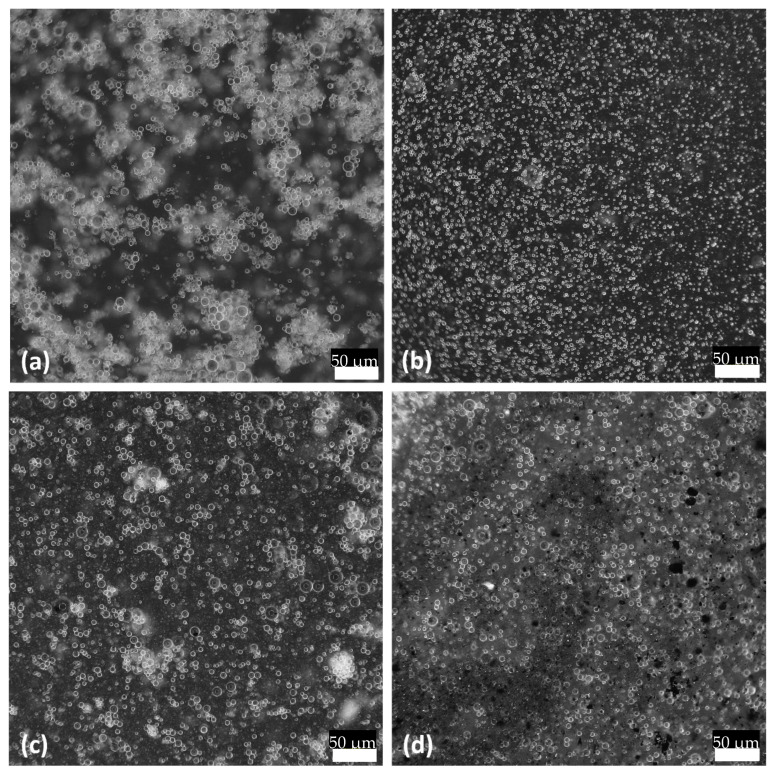
Optical microscope images of droplets in water dispersion of (**a**) TO/Tween 80, (**b**) TO/gel, (**c**) TO + 1 wt. % GNP/Tween 80, and (**d**) TO + 1 wt. % GNP/gel. These images were taken directly after 30 min of homogenization at 10,000 rpm.

**Figure 4 polymers-16-00909-f004:**
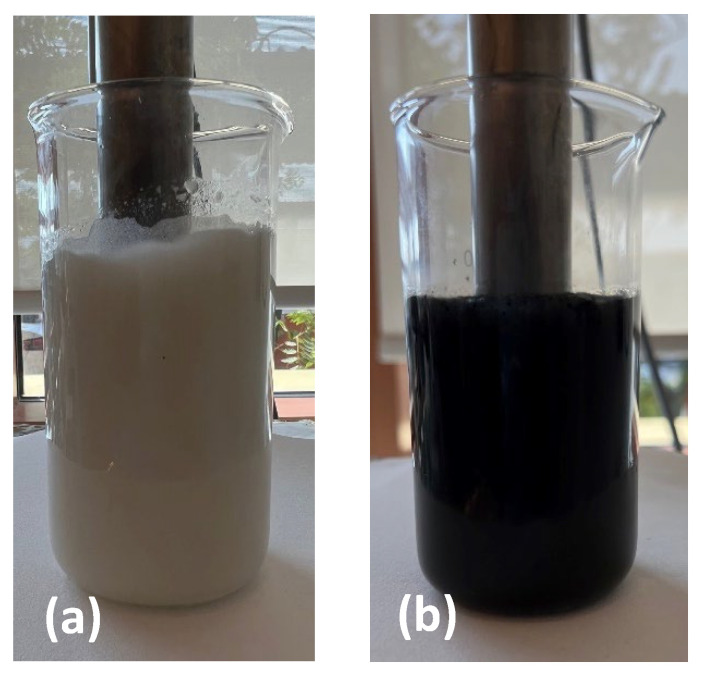
Images of the O/W emulsion during homogenization with (**a**) TO and gel emulsifier, and (**b**) GNP/TO and gel emulsifier.

**Figure 5 polymers-16-00909-f005:**
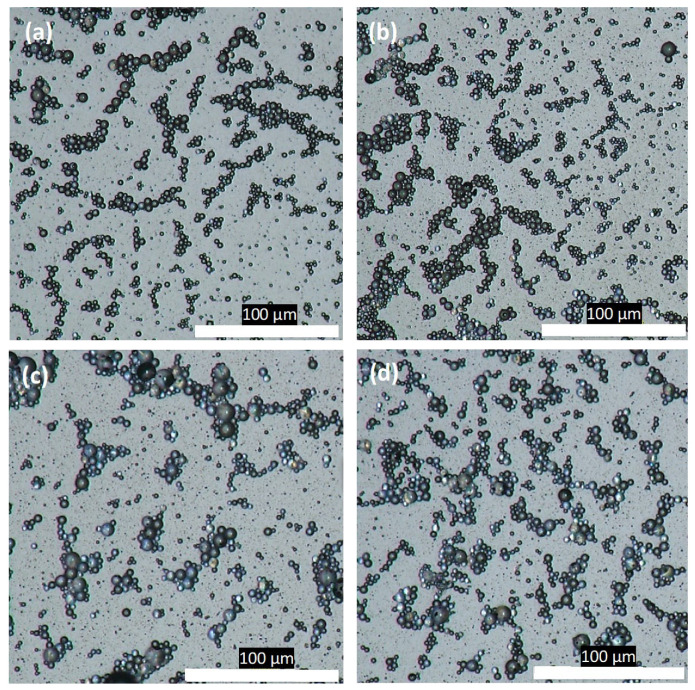
OM images for (**a**) Pure TO MCs, (**b**) 1% GNP/TO MCs, (**c**) 3% GNP/TO MCs, (**d**) 5% GNP/TO MCs. Images taken after filtration and drying of the MCs.

**Figure 6 polymers-16-00909-f006:**
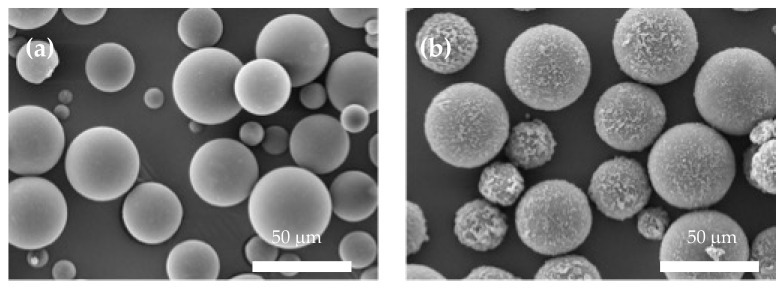
SEM images of (**a**) overall morphology of TO MC, (**b**) overall morphology of 1 wt. % GNP/TO MC. All images in the same conditions and coated with 5 nm of gold sputtering.

**Figure 7 polymers-16-00909-f007:**
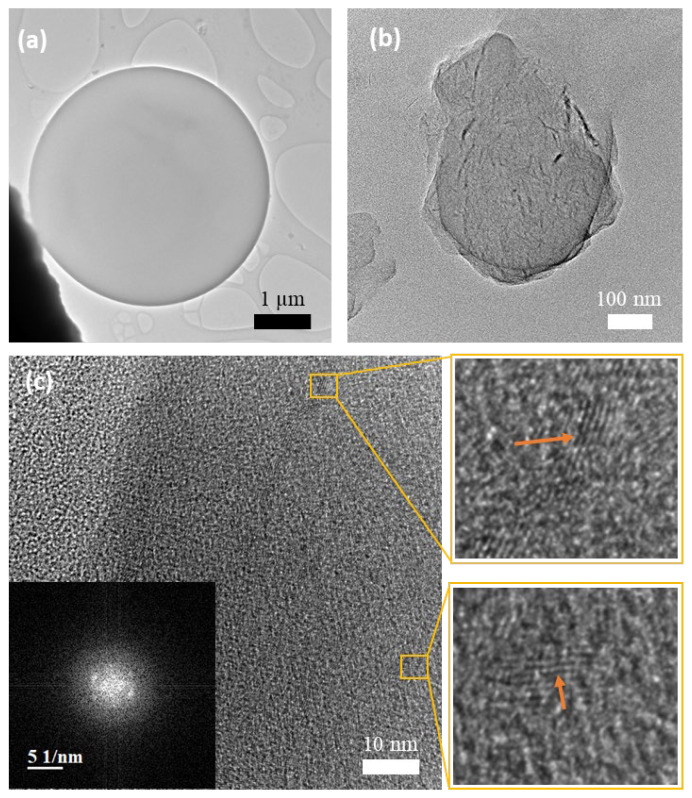
TEM images of the MCs. A bright TEM image of (**a**) an MC (without GNPs) and (**b**) GNP-incorporated MC. (**c**) High-magnification TEM image of an MC with GNP coating. High-magnification (zoomed in) images indicate the periodic graphene layers (indicated by the arrows).

**Figure 8 polymers-16-00909-f008:**
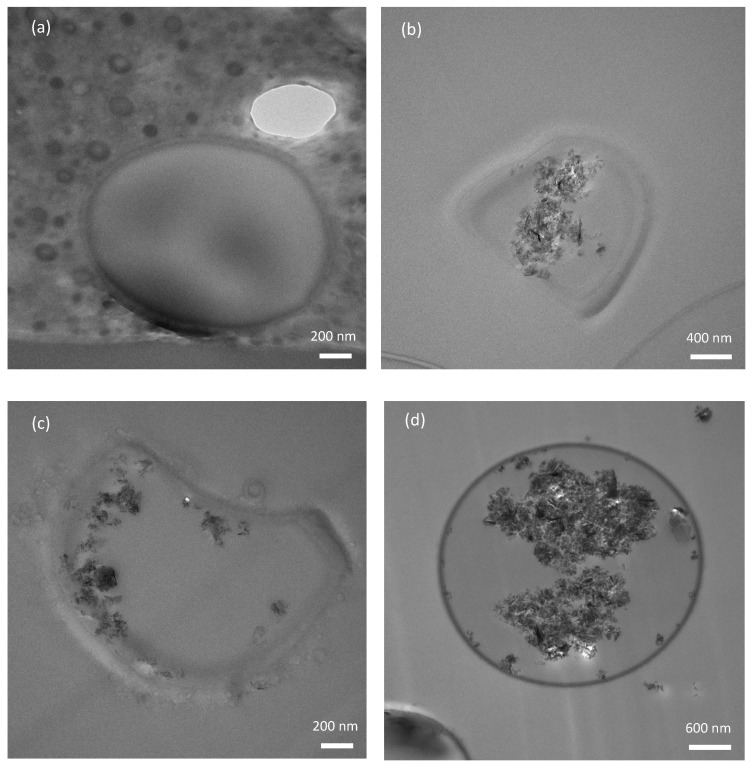
TEM images of the cross-sectional MCs sliced in the epoxy present an internal view of MCs. (**a**) Pure TO MC, (**b**) 1% GNP/TO MC, (**c**) 3% GNP/TO MC, and (**d**) 5% GNP/TO MC. The graphene species can be seen to be contained in the core of the microcapsules when the GNP is added in the microencapsulation process.

**Figure 9 polymers-16-00909-f009:**
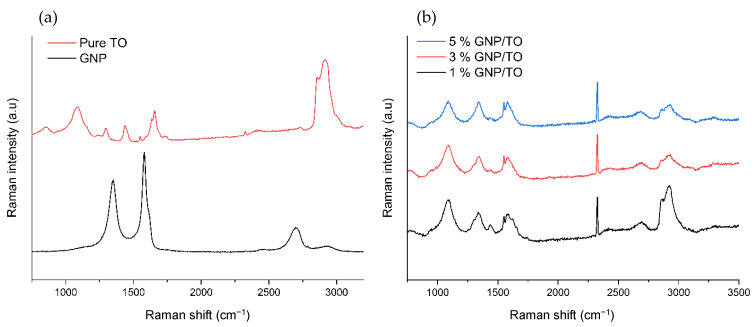
Raman spectra of (**a**) pure TO and pure GNP flakes, and (**b**) the core material content of 1%, 3%, and 5% GNP/TO MCs.

**Figure 10 polymers-16-00909-f010:**
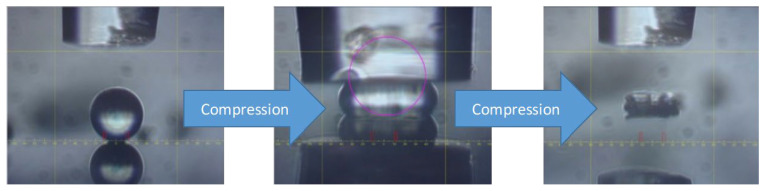
Individual pure TO MC before and after compression with a glass probe in the micromanipulation equipment.

**Figure 11 polymers-16-00909-f011:**
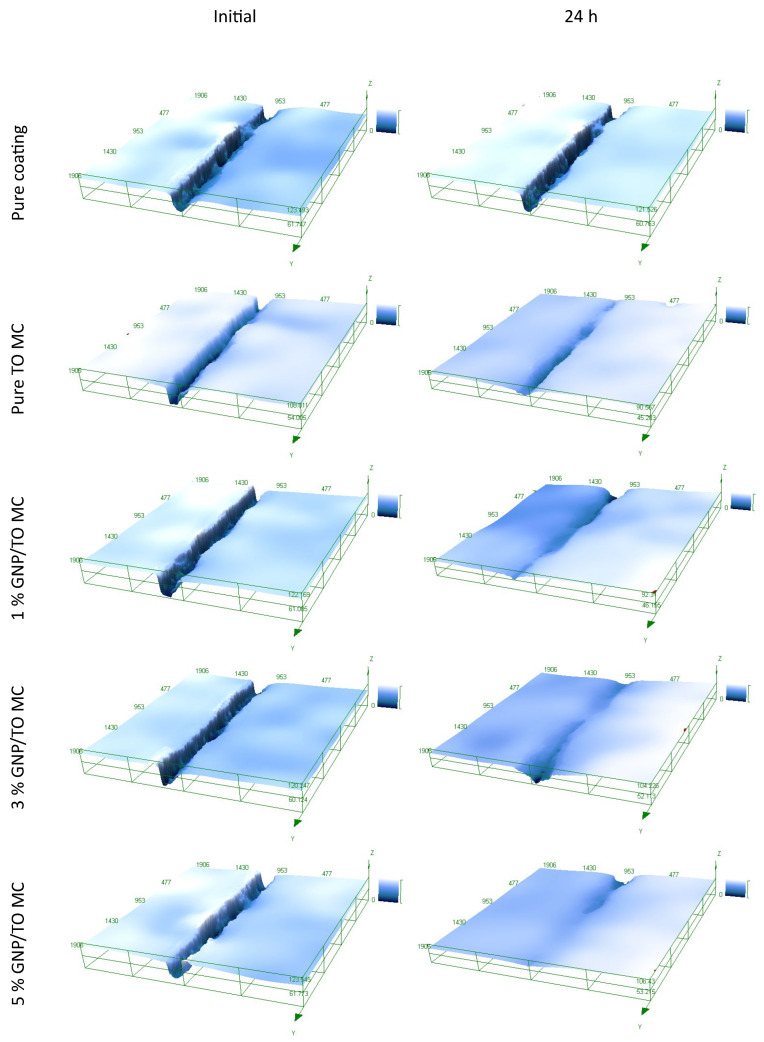
Three-dimensional optical microscopy images for the metallic substrates scratched initially, and after 24 h for the pure coating, the pure MCs, 1% GNP/TO, 3% GNP/TO, and 5% GNP/TO MCs embedded in the epoxy coating.

**Figure 12 polymers-16-00909-f012:**
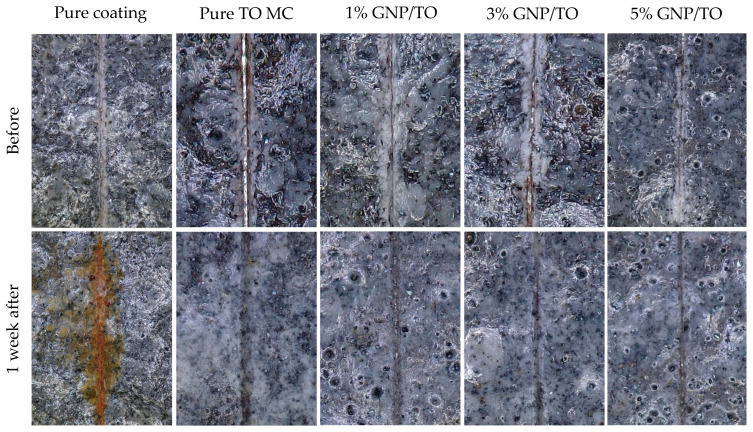
Steel substrates coated with the epoxy primer with and without the addition of the pure coating, pure tung oil MCs, and 1%, 3%, and 5% GNP/TO MCs.

**Table 1 polymers-16-00909-t001:** Droplet size distribution measured by Mastersizer.

Sample	Uniformity *	*D*_s_, μm	*D*_v_, μm	*D*_v_ (10), μm	*D*_v_ (50), μm	*D*_v_ (90), μm
Pure TO/gel	1.1	3.7	10.7	1.6	6.6	26.0
1% TO-GNP/gel	0.8	2.9	7.9	1.2	6.0	17.4
3% TO-GNP/gel	0.9	3.4	10.5	1.4	7.6	23.1
5% TO-GNP/gel	2.1	3.3	19.4	1.3	7.7	25.6
Pure TO/tween 80	0.5	5.7	11.6	2.8	10.6	21.6
1% TO-GNP/tween 80	0.8	5.2	13.5	2.2	10.7	28.8
3% TO-GNP/tween 80	1.0	4.6	14.3	2.0	10.0	33.6
5% TO-GNP/tween 80	0.7	6.4	15.6	2.8	12.8	32.6
5% TO-GNP/gel/ tween 80	2.4	1.9	12	0.82	4.3	23.9

* Uniformity—a measure of the absolute deviation from the median; *D*_s_ is the Surface Weighted Mean Diameter; *D*_v_ is the Volume Weighted Mean Diameter; *D*_v_ (10/50/90) is the size of particles at which 10/50/90% of the sample lies.

**Table 2 polymers-16-00909-t002:** Results of the micromanipulation of the pure, 1% GNP/TO MC, 3% GNP/TO MC, and 5% GNP/TO MC samples.

Sample	Diameter (D), µm	Displacement at Rupture (δ_r_), µm	Rupture Force (F_r_), mN	Apparent Toughness (T_c_), MPa	Apparent Young’s Modulus (E), MPa
Pure TO MC	9.7 ± 0.8	3.0 ± 0.7	0.25 ± 0.10	0.52 ± 0.30	28 ± 4
1% GNP/TO MC	15.2 ± 1.0	7.6 ± 0.5	0.76 ± 0.06	0.89 ± 0.10	39 ± 3
3% GNP/TO MC	12.9 ± 0.7	6.6 ± 0.4	0.63 ± 0.04	0.82 ± 0.07	40 ± 2
5% GNP/TO MC	11.2 ± 0.4	6.3 ± 0.2	0.66 ± 0.04	0.89 ± 0.06	38 ± 3

**Table 3 polymers-16-00909-t003:** Pull-off testing results for the pure coating, pure microcapsules, 1% GNP, 3% GNP, and the 5% GNP MCs embedded in the epoxy coating.

Sample	Adhesion Strength (MPa)
Pure coating only	12.82 ± 0.03
Pure TO MC	12.61± 0.05
1% GNP/TO MC	12.22 ± 0.07
3% GNP/TO MC	12.18 ± 0.06
5% GNP/TO MC	12.16 ± 0.04

**Table 4 polymers-16-00909-t004:** Healing efficiencies for the pure coating, pure MCs, 1% GNP/TO, 3% GNP/TO, and the 5% GNP/TO MCs embedded in the epoxy coating.

Sample	Healing Efficiency (%)
Pure coating only	3 ± 1
Pure TO MC	87± 1
1% GNP/TO MC	91 ± 2
3% GNP/TO MC	91 ± 1
5% GNP/TO MC	93 ± 2

## Data Availability

Data are contained within the article.

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
