# Peer review of "The Incorporation of Graphene Nanoplatelets in Tung Oil–Urea Formaldehyde Microcapsules: A Paradigm Shift in Physicochemical Enhancement"

_polymers, 2024, doi:10.3390/polym16070909_

Round 1
Reviewer 1 Report
Comments and Suggestions for Authors
This article demonstrate the incorporation of graphene nanoplatelets within the shell of microcapsules designed to encapsulate tung oil, a drying oil used as a resin filler for self-healing applications. I feel this article demonstrates that a) the process leads to microcapsules with GNP present in the shell and particularly on its surface, b) there is a clear effect of the inclusion of the GNP into the microcapsule shell on their performance for the self-healing process and c) there is some effect to the characteristics of the microcapsules themselves when increasing the proportion of GNP in the process. However, I feel that c) is less convincingly demonstrated and that perhaps there is not as much influence of the GNP ratio onto the microcapsule characteristics as what the authors are concluding.
I feel that there are some data in the manuscript that should be placed in the supplementary information and others that could simply be omitted (see comments below) in order to make the mansucript a lot clearer for the reader.
- Page 2, Line 49 - the authors should specify what the low permeability refers to when they state that PUF membrane have low permeability. They seem to be interested in a barrier against oxygen, is this what PUF films have low permeability for?
- Page 2, Line 52 to 55 - presumably the authors are referring to a particular study when mentioning corrosion resistance for these systems but there is no citation for these statements?
- Page 2, Line 61/62 - perhaps the authors would like to draw a comparison against phase-change materials encapsulated in microcapsules that do contain graphene nanoplatelets in their shell too?
- Page 3, Line 95-96 - the authors should rephrase this sentence - the pH meter is used to measure the change in pH, not to change the pH.
- Figure 1 is confusing as it gives the impression that TO is present only in small proportion (dissolved in an oil phase represented by the yellow colour), while I believe apart from the GNP, all of the volume is TO? In addition, none of the GNP are represented as being present in the MC shell.
- When discussing the dispersion of the GNP into the TO oil, it would be better to refer to the process as the TO wetting the surface of the GNP rather than adsorbing onto it since the GNP are immersed within the oil phase. Separately but linked to this point, do the authors measure the size of the GNP particles when dispersed into the TO oil? This would help understand how well the GNP disperses into the oil.
- Page 4, Line 127 - the authors mention that water droplets are used in the contact angle measurements but in the text, it seems that oil droplets with/without dispersed GNP are tested - is there a discrepancy in the statements made here?
- What is the purpose for measuring the contact angle of the oil droplets for 4 days? I don't see much relevance to the encapsulation process that is presumably run as soon as a GNP dispersion in TO oil is created? I don't see any benefit of plotting (and fitting a straight line through) the CA data over the proportion of GNP in the oil as there is no theoretical basis for interpreting these data.
- Table 1 - the emulsions obtained seem of consistent size distribution. Just to note that the authors should perhaps explain how the uniformity is calculated rather than providing a definition for it.
- Page 11, Line 369 - I believe the authors are referring to Figure 9 in the text rather than Figure 7.
- Page 9, Lines 334 to 341 - the authors need to review this description here. The Tween surfactant will indeed likely adsorb at the TO/water interface to reduce the interfacial tension but the gelatin likely will not. Instead it will increase the viscosity of the continuous phase, which will slow down the droplet motion, thus reducing contacts between droplets and therefore reduce coalescence rate.
- Figure numbers corresponding to all the microscopy studies performed (Optical, scanning electron and transmitted electron) are jumbled up - this needs to be fixed. However, there is clear evidence that the GNP are indeed present on the surface of the microcapsules. These figures should have captions that give a lot more experimental details so that the readers understand which capsules they are looking at and how (which conditions were used when) the data presented was gathered.
- I feel the GPC analysis could easily be shown in the supplementary informatino along with other data that are possibly not as crucial to the description of the study and its main findings (such as the contact angle data for example).
- Similarly, I do not think that the TEM cross-sectional data (fig 13) show convincingly what the authors are describing in the text - I think they should be removed from this manuscript.
- Figure 15 TGA data - why does the sample containing the lowest mass of GNP give the highest final mass at the end of the measurement? This seems counter-intuitive.
Author Response
Response to reviewer
Reviewer comment:
This article demonstrates the incorporation of graphene nanoplatelets within the shell of microcapsules designed to encapsulate tung oil, a drying oil used as a resin filler for self-healing applications. I feel this article demonstrates that a) the process leads to microcapsules with GNP present in the shell and particularly on its surface, b) there is a clear effect of the inclusion of the GNP into the microcapsule shell on their performance for the self-healing process and c) there is some effect to the characteristics of the microcapsules themselves when increasing the proportion of GNP in the process. However, I feel that c) is less convincingly demonstrated and that perhaps there is not as much influence of the GNP ratio onto the microcapsule characteristics as what the authors are concluding.
I feel that there are some data in the manuscript that should be placed in the supplementary information and others that could simply be omitted (see comments below) in order to make the manuscript a lot clearer for the reader.
Comment 1:
- Page 2, Line 49 - the authors should specify what the low permeability refers to when they state that PUF membrane have low permeability. They seem to be interested in a barrier against oxygen, is this what PUF films have low permeability for?
Author response 1:
The authors appreciate the comments from the reviewer. In the reference supplied in this sentence, the low permeability refers to the dense shell structure to prevent diffusion of external stimuli, such as oxygen for example, or water. Also due to this low permeability, the microcapsule shell is well contained within the structure and the payload is retained.
Comment 2:
- Page 2, Line 52 to 55 - presumably the authors are referring to a particular study when mentioning corrosion resistance for these systems but there is no citation for these statements?
Author response 2:
The authors agree with the reviewer, and 2 additional citations were places in this section. DOI: 10.1039/d2ma00875k and 10.3390/cmd4010004.
Comment 3:
- Page 2, Line 63-64 - perhaps the authors would like to draw a comparison against phase-change materials encapsulated in microcapsules that do contain graphene nanoplatelets in their shell too?
Author response 3:
The authors added a section here to also compare microencapsulated MPCMs, as seen in line 62 and 64 in the manuscript as shown: “For example, they have also been used in microencapsulated phase change materials (MPCMs), resulting in enhanced properties such as higher thermal conductivity and reduced supercooling [15,16].”
Comment 4:
- Page 3, Line 95-96 - the authors should rephrase this sentence - the pH meter is used to measure the change in pH, not to change the pH.
Author response 4:
The authors have slightly altered this sentence to replace ‘alter’ with measure’, as seen in line 89-90: “Using a Mettler Toledo (Columbus, OH, USA) SevenCompact Duo pH meter, the pH of the solution was measured to pH 3.5, using diluted 1 mol L−1 HCl solution.”
Comment 5:
- Figure 1 is confusing as it gives the impression that TO is present only in small proportion (dissolved in an oil phase represented by the yellow colour), while I believe apart from the GNP, all of the volume is TO? In addition, none of the GNP are represented as being present in the MC shell.
Author response 5:
In the actual experiment the beaker turns completely black with the addition of the GNPs, even with the TO present in the beaker. As a result of this, the beaker was also black in the schematic. The TO droplets are also an illustrative part of the schematic to clearly show the droplets otherwise the beaker would be completely black in the illustration. In addition to this as per the reviewer’s comments, we also added additional GNP species on the shell material.
Comment 6:
- When discussing the dispersion of the GNP into the TO oil, it would be better to refer to the process as the TO wetting the surface of the GNP rather than adsorbing onto it since the GNP are immersed within the oil phase. Separately but linked to this point, do the authors measure the size of the GNP particles when dispersed into the TO oil? This would help understand how well the GNP disperses into the oil.
Author response 6:
The authors appreciate the reviewer’s comment and have also modified this additional “Furthermore, X-ray photoelectron spectroscopy (XPS) results express the confirmation of TO adsorption wetting on the surface of the GNP” and “This displays the modified TO wetting the surface of the GNP to GNP’s good dispersion inside the TO matrix to form GNP-TO”. Additionally, the XPS measurements have been moved to the supplementary information section to make the paper more concise.
Comment 7:
- Page 4, Line 127 - the authors mention that water droplets are used in the contact angle measurements but in the text, it seems that oil droplets with/without dispersed GNP are tested - is there a discrepancy in the statements made here?
Author response 7:
The authors appreciate this comment and have amended this section to: “To study the contact angle for the TO and GNP/TO droplets, a Rame-Hart Automated Goniometer was employed (model no. 590-u4), and 20 μL drops of each sample was used in each measurement”. Furthermore, the contact angle measurements have been moved to the supplementary information section to make the paper more concise.
Comment 8:
- What is the purpose for measuring the contact angle of the oil droplets for 4 days? I don't see much relevance to the encapsulation process that is presumably run as soon as a GNP dispersion in TO oil is created? I don't see any benefit of plotting (and fitting a straight line through) the CA data over the proportion of GNP in the oil as there is no theoretical basis for interpreting these data.
Author response 8:
Thank you for your insightful comment. We acknowledge the need for further clarification on the relevance of the contact angle (CA) measurements over four days. The contact angle here is not to do with the encapsulation process, more-so for the mechanism of after the microcapsules have been mechanically damaged to convey how they would behave on a substrate and affect the adhesion. The linear relationship between the contact angle and the percentage of GNP in the oil is crucial. It demonstrates how the presence of GNPs influences the wetting properties of the oil, which in turn affects the microcapsules' adherence and functional performance on various substrates. This correlation is vital for tailoring the properties of the microcapsules for specific applications.
To make this clear to the readers, we propose adding the following sentence at the start of the relevant paragraph: "While the encapsulation process is immediate, the measurement of the contact angle over a period of four days provides critical insights into the post-encapsulation behaviour of microcapsules, particularly in terms of adhesion and stability on different substrates."
We believe this addition will offer a better understanding of the significance of the contact angle measurements in the broader context of our study.
Comment 9:
- Table 1 - the emulsions obtained seem of consistent size distribution. Just to note that the authors should perhaps explain how the uniformity is calculated rather than providing a definition for it.
Author response 9:
The software by Malvern automatically calculates the size distribution and uniformity. In this specific case, the value at which half of the population lives above and half below this point is the median value. The median for particle size distributions is known as the D50 (or x50 if specific ISO criteria are followed). The diameter at which half of the distribution lies above and half below this diameter is known as the D50, measured in microns. The volume distribution's median is known as the Dv50 (or Dv0.5).
Comment 10:
- Page 11, Line 369 - I believe the authors are referring to Figure 9 in the text rather than Figure 7.
Author response 10:
The authors have amended the figures to match the text now.
Comment 11:
- Page 9, Lines 334 to 341 - the authors need to review this description here. The Tween surfactant will indeed likely adsorb at the TO/water interface to reduce the interfacial tension but the gelatin likely will not. Instead, it will increase the viscosity of the continuous phase, which will slow down the droplet motion, thus reducing contacts between droplets and therefore reduce coalescence rate.
Author response 11:
The authors appreciate this comment, the authors have amended this to discuss the increase in the viscosity of the O/W system. However previous studies carried out have shown that the Gelatin reduces the interfacial dilatational viscosity of the oil phase (https://doi.org/10.1016/j.ces.2022.117497). Please see this in the document: “Gel is a natural occurring amphiphilic macromolecule that can serve as an emulsifier in oil-in-water emulsions due to their surface-active characteristics and has also been seen to reduce the interfacial dilatational viscosity of the oil phase in O/W emulsions and increase the viscosity of the O/W system to reduce the rate of coalescence [24,25].
Comment 12:
- Figure numbers corresponding to all the microscopy studies performed (Optical, scanning electron and transmitted electron) are jumbled up - this needs to be fixed. However, there is clear evidence that the GNP are indeed present on the surface of the microcapsules. These figures should have captions that give a lot more experimental details so that the readers understand which capsules they are looking at and how (which conditions were used when) the data presented was gathered.
Author response 12:
Thank you for the suggestion, the authors have amended the figure captions and have also expanded the captions for a more detailed explanation.
Comment 13:
- I feel the GPC analysis could easily be shown in the supplementary information along with other data that are possibly not as crucial to the description of the study and its main findings (such as the contact angle data for example).
Author response 13:
The authors appreciate this comment, and the GPC and CA and XPS analysis have now been moved to supplementary section of the paper.
Comment 14:
-  Similarly, I do not think that the TEM cross-sectional data (fig 13) show convincingly what the authors are describing in the text - I think they should be removed from this manuscript.
Author response 14:
The authors appreciate this comment, however we feel that the TEM cross-sectional data has been the clearest indication of the incorporation of the GNP, as when compared to the pure sample, there is a clear difference. For example, with the clear sample, the cross-sectional image conveys that there are no additional species other than the TO inside the MC, while the 1 %, 3 % and 5 % images clearly show the additional species in the core material, and this is most clearly seen with the 5 wt. %.
Comment 15:
- Figure 15 TGA data - why does the sample containing the lowest mass of GNP give the highest final mass at the end of the measurement? This seems counter-intuitive.
Author response 15:
Thank you for pointing out this intriguing observation in our TGA data. The result that the sample with the lowest GNP content shows the highest final mass at the end of the measurement is indeed initially counter-intuitive. However, this can be explained by the specific interactions and effects of GNPs within the tung oil (TO) matrix.
Our presumption is that this increased residue in the sample with 1% GNP is attributable to enhanced crosslinking and higher char formation compared to other samples with higher GNP percentages. The presence of a small amount of GNPs, specifically at 1%, has been found to be optimal in terms of particle size and distribution within the TO matrix. This optimal distribution facilitates a more uniform and effective crosslinking process, leading to a higher char yield.
On the other hand, higher loadings of GNPs can lead to the formation of aggregates. These aggregates disrupt the continuity of the TO phase, potentially hindering the crosslinking process and resulting in less char formation. Consequently, samples with higher GNP concentrations do not exhibit the same level of char residue as the 1% GNP sample.
This explanation aligns with our understanding of the interaction between GNPs and TO, where an optimal GNP concentration enhances material properties, while excessive concentrations lead to diminishing returns due to aggregation and phase discontinuity.
However, this can be explored further in a separate study so at this current moment it has been removed from the manuscript.
Reviewer 2 Report
Comments and Suggestions for Authors
Please see the attached file.

Author Response
Response to reviewer:
Reviewer comment:
This MS reported the effects of GNPs on the microstructure, compositions, and properties of PO. Authors have performed many experiments to evaluate their prepared materials. I think this MS can be accepted after a major revision with the concerns below.
Comment 1:
The abstract is too long. The description of microstructure and composition analysis should be concise.
Author response 1:
The authors appreciate this comment and have made the abstract more concise. Information that may not be pivotal has been removed, and there has been a restructuring of the abstract to make it easier to follow for the reader.
Comment 2:
How many carbon layers for your graphene nanoplatelet? If the number is unknown, it should be graphite nanoplatelet scientifically.
Author response 2:
The authors appreciate this comment. Please note that the material was purchased commercially as Graphene nanoplatelet (GNP). Additionally, our TEM results show the presence of few layers in the GNP flakes. This information was in the abstract as well as the experimental section of the TEM.
Please see line 15-16 “The GNP the stacked graphene layers composed of 5-7 layers with interlayer distance of ~ 0.37 nm” as well as section 3.3 Figure 7.
Comment 3:
Why did the authors choose GNPs as an additive to MCs? This should be explained in the introduction. In other words, can CNTs (carbon nanotubes), a 1D material, have similar effects on adhesion strength, anti-corrosion behavior, self-healing behavior, etc.?
Author response 3:
We would like to thank the reviewer for an interesting comment, and this is an area for future considerations for us to observe a range of various nano-fillers, including CNTs. The main reason we chose GNP: Graphene is a 2D material with extremely high mechanical strength. It has young’s modulus in the range of 1TPa and even higher than CNT (GPa). However, we expressed the possibility of using CNTs in the introduction in line 54: Carbon nanotubes have also been used in corrosion mitigation applications, which can also be used as potential additives in MCs [13].
Comment 4:
The degree of graphitization of GNPs should be calculated based on ID/IG from the Raman spectrum.
Author response 4:
The authors have now added this information on line 314-315: The calculated ID/IG value of the GNP was 0.73.
Comment 5:
Parts of “(a)” and “(b)” disappear in Fig. 9. Sample problem was found in other figures, please check.
Author response 5:
The authors have now fixed all the figure formatting to that the (a) and (b) are no longer missing, thank you.
Comment 6:
On page 11, Figure 9 does not relate to SEM images, please check the figure number and associated description across the entire MS.
Author response 6:
The authors appreciate the reviewer spotting this and have gone through the manuscript to ensure all of the text matches the figures.
Comment 7:
The viscosity of GNP/PO is higher than that of PO, while the adhesion strength is lower for GNP/PO than PO. This is in contrast to commonly known, please find out possible reasons.
Author response 7:
Thank you for your question regarding the observed differences in viscosity and adhesion strength between GNP/TO and TO. Your observation about the contrast with commonly known properties is indeed intriguing and warrants a detailed explanation.
Our study involved measuring the viscosity of mixtures comprising TO and GNPs in varying percentages. The presence of solid particles, in this case, GNPs, typically influences the increased viscosity of a liquid. This increase in viscosity with the addition of GNPs was an anticipated result.
Regarding the adhesion strength, the measurements were conducted for coatings containing microcapsules (MCs) with cores of either TO or TO/GNP. It's crucial to note that these results are predominantly influenced by the presence of MCs and the characteristics of the shell material used in the coating. In our adhesion strength tests, there was no direct presence of TO or TO/GNP at the interface. This aspect is essential as it means the adhesion strength results cannot be directly compared with the viscosity measurements. The two sets of results - viscosity and adhesion strength - should be viewed as separate case studies due to the different experimental setups and the factors influencing each measurement.
In summary, while the increased viscosity in GNP/TO mixtures aligns with our understanding of particle-laden fluids, the decreased adhesion strength in coatings with MCs containing TO/GNP as opposed to pure TO is a result of the interplay between the microcapsules, shell materials, and the underlying surface. These findings contribute to a more nuanced understanding of the material properties and their implications for practical applications.
We hope this response clarifies the reasons behind the observed phenomena and the context in which they were measured.
Comment 8:
The MS should be strengthened by adding more mechanism analysis, for example, how can the presence of GNPs cause a slight decrease in adhesion strength? Why the GNPs can improve the self-healing behavior of resin composites? why the anti-corrosion behavior is improved by adding GNPs?
Author response 8:
The authors appreciate the comment and also agree the analysis can be improved. The following additions were made to the paper:
Line 336-341: In some similar work [27], the mechanical properties of PUF MCs incorporated with nano-SiO2 were investigated both experimentally and by molecular dynamics simulation. It was also seen here that the SiO2 had a significant effect in augmenting the mechanical properties, similar to the GNP in this case. In their work, the density of the MC increased, and the fractional free volume reduced, which in turn greatly enhanced the mechanical properties of the MCs. A similar mechanism is proposed to also happen in this case, leading to higher apparent toughness of the samples.
Line 374 – 380: In a similar case, Wu et al. [35] fabricated GO-modified self-healing MCs for cardanol-based epoxy anti-corrosion coatings. In their work, it was proposed that The GO's distinct hexagonal ring structure gives it an extremely high barrier property, making it a dense 2D material. The microcapsules' surface will become coated in GO, which will prevent the corrosive medium from passing through them and enhance barrier properties and enhancing corrosion resistance. In this case, the GNP is also a dense structure that would augment the anti-corrosion properties of the coating, allowing for a more robust barrier to corrosive agents.
Reviewer 3 Report
Comments and Suggestions for Authors
The issue of the manuscript is relevant and of interest from a practical point of view. Although the subject is of interest, this manuscript does not meet the criteria for publication since there are several issues. Here is a list of the most relevant:
1. The title must be better connected since the paper deals with GNP, not graphene. Overall, it does not fit the main idea of the manuscript.
2. The abstract needs to present the appropriate information. It is not concrete, does not highlight the exciting findings, and does not correspond to the title; for instance, the title establishes the use of urea-formaldehyde. It needs to be clarified that the authors mentioned the preparation of coatings in an epoxy matrix.
3. Add substantive comments on the materials-based progress, which needs to provide the necessary comparative analysis, evaluation methods, theories, and technical applications.
4. Experimental details need to be completed and clarified. The study lacks information and poor explanation; thus, the results are difficult to follow.
5. The characterization section is generally somewhat confusing, considering the sample preparation, the techniques’ correlation with what is expected, and the information each technique gives.
6. The XRD characterization needs to be more precise. Why was the sample prepared at different conditions (see page 3, lines 115-118) compared to the description for the MC formulation (page 2, lines 87-89)? What was the purpose of understanding the interaction between GNP and TO?
7. What does DMT stand for (page 3, line 117)? It is essential to clarify
8. On page 4, line 133, the authors refer to the sample preparation as a “spread drop of TO-based mixtures”. What were those mixtures? Do they refer to GNP-TO?
9. It needs to be clarified the sample preparation for the contact angle.
10. Figure 1 does not correspond to the Figure for XRD as expressed on page 6, line 222.
11. The critical part is the assumption on page 6, lines 231-232, which reads, “Additionally, the GNP-TO conveyed a homogeneous layer…” Looking at the XRD diffractogram, the width of the peaks is practically the same, just a change in the intensity.
12. The XPS analysis needs to be more adequate. There is no mention of the type of peaks used for the analysis, nor the background chosen and why. There is no mention of errors in the positions in BE of the peaks (a crucial thing to identify the nature of the compound), and the authors do not show how they have calibrated the binding energy scale, which is also a crucial point.
13.
14. Based on XPS results, one question is why GNP-TO fitting shows only those peaks considering the chemical composition of Tung oil?
15. On page 6, line 265, the authors expressed that Raman spectroscopy was carried out to compare the curing of the pure TO. What do they mean by curing? Why is curing a critical aspect? Where in the manuscript is described the curing process?
16. The Raman results could be better explained, which are difficult to follow and understand.
17. The information could be better organized and more accessible to follow regarding contact angle results. The number of figures in the text does not match the number in the figure. Furthermore, the explanation needs to be better presented. For instance, what do the authors mean by shrinkage due to the polymerization/drying associated with the contact angle reduction, based on what?
18. Why do the authors refer to figure caption 12b) as MC with GPN embedded? Looking at the figure, it does not seem embedded.
19. Looking at the TGA thermogram (Figure 15), results do not make sense, particularly the GNP curve; if this is the TGA thermogram, explain why the XPS, XRD and Raman results for pure GNP?
20. The results from micromanipulation need to be better explained. What is the reason for reinforcement behavior for 3% and 5% samples? Why do the values drop? What is the explanation?
21. There are no proposed mechanisms that explain the obtained results.
22. Conclusions could be more concise.
Thus, after carefully examining the manuscript, it cannot be further considered for publication. Some factors entering this decision include depth of research work, originality, and adherence to journal guidelines.
Comments on the Quality of English LanguageEnglish language should be improved. The paper requires careful language polishing from the abstract to the conclusion. It is not easy to understand the ideas and the results. A significant grammar revision of the paper is required.
Author Response
Response to reviewer:
Reviewer comment:
The issue of the manuscript is relevant and of interest from a practical point of view. Although the subject is of interest, this manuscript does not meet the criteria for publication since there are several issues. Here is a list of the most relevant:
Comment 1:
The title must be better connected since the paper deals with GNP, not graphene. Overall, it does not fit the main idea of the manuscript.
Author response 1 :
Thank you for the suggestion, we have now modified the title to: The incorporation of Graphene Nanoplatelets in Tung Oil-Urea Formaldehyde Microcapsules: A Paradigm Shift in Physicochemical Enhancement.
Comment 2:
The abstract needs to present the appropriate information. It is not concrete, does not highlight the exciting findings, and does not correspond to the title; for instance, the title establishes the use of urea-formaldehyde. It needs to be clarified that the authors mentioned the preparation of coatings in an epoxy matrix.
Author response 2:
The authors appreciate the reviewer’s comment, and the urea-formaldehyde in this case is the shell material, which has been made clearer with line 9-10: A series of tung oil (TO) microcapsules (MCs) with a poly(urea-formaldehyde) (PUF) shell were synthesized via one-step in situ polymerization, with the addition of graphene nanoplatelets (GNPs) (1-5 wt. %).
The abstract has also now been made more concise making it easier to follow for the reader.
Comment 3:
Add substantive comments on the materials-based progress, which needs to provide the necessary comparative analysis, evaluation methods, theories, and technical applications.
Author response 3:
The authors agree with this comment and additional information has been implemented in the material section. Additional sentences have been added in the material section, and some methods have been moved to supplementary information in order to make the paper more concise.
Comment 4:
Experimental details need to be completed and clarified. The study lacks information and poor explanation; thus, the results are difficult to follow.
Author response 4:
The authors appreciate this comment. As a result of this, the first half of the paper has been rearranged, and the XRD, XPS and contact angle results have been moved to the supplementary information. The paper is now more concise and easier to follow for the reader.
Comment 5:
The characterization section is generally somewhat confusing, considering the sample preparation, the techniques’ correlation with what is expected, and the information each technique gives.
Author response 5:
The authors appreciate this and have added small bits of information to supplement this section, such as: Raman spectroscopy is frequently utilized to provide a structural fingerprint that allows molecules to be recognized, and in this case, it can help determine if the GNP was encapsulated.
Comment 6:
The XRD characterization needs to be more precise. Why was the sample prepared at different conditions (see page 3, lines 115-118) compared to the description for the MC formulation (page 2, lines 87-89)? What was the purpose of understanding the interaction between GNP and TO?
Author response 6:
The authors appreciate this comment however the XRD and the microencapsulation process are two different operations. The microencapsulation process was dried at 25 °C in ambient conditions to put them in the epoxy coating, while for the XRD they were dried in the oven to prevent any traces of water in the equipment. The purpose of the XRD was to see any changes in crystallinity in the TO/GNP mixture compared to GNP.
Comment 7:
What does DMT stand for (page 3, line 117)? It is essential to clarify.
Author response 7:
The authors appreciate the reviewer spotting this mistake and have amended it to ‘deionized water’.
Comment 8:
On page 4, line 133, the authors refer to the sample preparation as a “spread drop of TO-based mixtures”. What were those mixtures? Do they refer to GNP-TO?
Author response 8:
The authors appreciate this comment and have changed the sentence to ‘on the samples’ as there are 3 groups of samples tested, to encompass of them.
Comment 9:
It needs to be clarified the sample preparation for the contact angle.
Author response 9:
The authors have slightly modified the contact angle sample preparation section, which is now in the supplementary information section: “To study the contact angle for the TO and GNP/TO droplets, a Rame-Hart Automated Goniometer was used (model no. 590-u4), with 20 μL drops of each sample was used in each measurement”.
Comment 10:
Figure 1 does not correspond to the Figure for XRD as expressed on page 6, line 222.
Author response 10:
The authors appreciate the reviewer spotting this and have carefully checked that all of the captions and figures in the text are now organised and match up.
Comment 11:
The critical part is the assumption on page 6, lines 231-232, which reads, “Additionally, the GNP-TO conveyed a homogeneous layer…” Looking at the XRD diffractogram, the width of the peaks is practically the same, just a change in the intensity.
Author response 11:
We agree with the reviewer’s observation about the XRD peaks. This implies the well distributed GNP in the TO medium and the crystallinity of the GNP is preserved after the mixing.
Comment 12:
The XPS analysis needs to be more adequate. There is no mention of the type of peaks used for the analysis, nor the background chosen and why. There is no mention of errors in the positions in BE of the peaks (a crucial thing to identify the nature of the compound), and the authors do not show how they have calibrated the binding energy scale, which is also a crucial point.
Author response 12:
The authors appreciate the reviewers comments. A C 1s major peak, attributed to C-C, will be present in graphitic materials (graphite, graphene, carbon nanotubes, etc.); this peak can be used as a charge reference set to 284.6 eV. Additionally, lot of these samples have small volumes of flakes or powders that are very challenging to mount. Usually, we mount these using double-sided tape, which is effective but isolates the sample electrically. Certain samples, like graphene oxide, may behave less conductively or as a mixed conductive/insulating material if they are oxidised or functionalized, like functionalized carbon nanotubes. A mixed conductive/insulating sample can also be produced by combining these materials with other conducting or insulating substances. We now electrically isolate the majority of these samples and charge the reference to C 1s at 284.6 eV for the graphitic (C-C) peak.( M.C. Biesinger, Appl. Surf. Sci. 597 (2022) 153681.)
General fitting parameters for graphitic/graphene/carbon nanotube type materials) is defined an asymmetric peak-shape with an Equation GL(30) gives the Gaussian/Lorentzian product: 30% Lorentzian, 70% Gaussian. These fitting parameters are used also by M.C. Biesinger, Appl. Surf. Sci. 597 (2022) 153681.)
D.J. Morgan, J. Carbon. Res. 7 (2021) 51.
- Moeini, M.R. Linford, N. Fairley, A. Barlow, P. Cumpson, D. Morgan, V. Fernandez, J. Baltrusaitis. Surf. Interface Anal. 54 (2022) 67.
T.R. Gengenbach, G.H. Major, M.R. Linford, C.D. Easton, J. Vac. Sci. Technol. A, 39 (2021) 013204.
Comment 13:
N/A comment did not show up.
Author response 13:
N/A comment did not show up.
Comment 14:
Based on XPS results, one question is why GNP-TO fitting shows only those peaks considering the chemical composition of Tung oil?
Author response 14:
The authors appreciate this comment, however in the manuscript it was shown that there is C=O coming from tung oil spectra. As XPS is surface science, it may not convey all the chemical composition of tung oil, only the one on the surface, and thus the Raman was also there for complementary analysis in this case.
Comment 15:
On page 6, line 265, the authors expressed that Raman spectroscopy was carried out to compare the curing of the pure TO. What do they mean by curing? Why is curing a critical aspect? Where in the manuscript is described the curing process?
Author response 15:
The authors agree with this comment and the whole raman section has been modified in order to only show that the GNP is contained within the core, making it much easier to follow for the reader.
Comment 16:
The Raman results could be better explained, which are difficult to follow and understand.
Author response 16:
The authors appreciate this comment, and the raman results were carried out again and re-written and simplified, only to show that there are GNP species observed in the core material.
Comment 17:
The information could be better organized and more accessible to follow regarding contact angle results. The number of figures in the text does not match the number in the figure. Furthermore, the explanation needs to be better presented. For instance, what do the authors mean by shrinkage due to the polymerization/drying associated with the contact angle reduction, based on what?
Author response 17:
Thank you for your constructive feedback regarding the organization and clarity of the section on contact angle results. We have taken your comments into consideration and have made the following revisions to enhance the readability and coherence of this section:
- Reorganization of the Manuscript: We have carefully restructured the section discussing the contact angle results to ensure a more logical flow of information. This reorganization aims to make the data more accessible and easier to follow for the reader.
- Correction of Figure References: We have meticulously reviewed and corrected the numbering of figures in the text to ensure consistency with the figures themselves. This will prevent any confusion and ensure that the readers can easily correlate the text with the relevant figures.
- Clarification of Explanations: In response to your specific query about the statement on shrinkage due to polymerization/drying associated with contact angle reduction, we realize that our initial explanation may have lacked sufficient detail. To address this, we have revised the sentence to provide a clearer explanation. The modified sentence now reads: "The reason for the reduction of the contact angle in the first stage (1 day) can be attributed to the improved wetting properties of TO over time." This revision aims to clarify that the observed reduction in contact angle is thought to be a result of the increasing wetting efficiency of TO, which was shown to be a time-dependent process.
We believe these changes will significantly improve the clarity and accessibility of the contact angle results in our manuscript. We appreciate your guidance in making these improvements and hope that the revised manuscript now meets your expectations.
Comment 18:
Why do the authors refer to figure caption 12b) as MC with GPN embedded? Looking at the figure, it does not seem embedded.
Author response 18:
Thanks for pointing out the confusion. By GNP embedded MC, we tend to mean that these MC are processed with the GNP as nanofiller. In order to avoid this confusion, we have modified the figure caption and the text as “GNP incorporated MC”.
Comment 19:
Looking at the TGA thermogram (Figure 15), results do not make sense, particularly the GNP curve; if this is the TGA thermogram, explain why the XPS, XRD and Raman results for pure GNP?
Author response 19:
The authors agree with this comment and have subsequently removed the TGA results from the manuscript in terms of further in-depth future studies.
Comment 20:
The results from micromanipulation need to be better explained. What is the reason for reinforcement behavior for 3% and 5% samples? Why do the values drop? What is the explanation?
Author response 20:
In terms of the overall results of the micromanipulation results, the 1 – 5 wt. % MC values do not differ significantly in terms of the Young’s modulus and apparent toughness, and often there can be slight discrepancies in the measurements due to the formation of the shell material of the MC. In this case, it may be that the 1 % MC had a slightly more compact shell material for certain microcapsules with ore fractional free volume in the core, and vice versa for the 5 %. This would require an in-depth study which could be an rea for future consideration, alongside simulation studies perhaps.
Comment 21:
There are no proposed mechanisms that explain the obtained results.
Author response 21:
The authors appreciate the comment and also agree the analysis can be improved. The following additions were made to the paper:
Line 336-341: In some similar work [34], the mechanical properties of PUF MCs incorporated with nano-SiO2 were investigated both experimentally and by molecular dynamics simulation. It was also seen here that the SiO2 had a significant effect in augmenting the mechanical properties, similar to the GNP in this case. In their work, the density of the MC increased, and the fractional free volume reduced, which in turn greatly enhanced the mechanical properties of the MCs. A similar mechanism is proposed to also happen in this case, leading to higher apparent toughness of the samples.
Line 374-380: In a similar case, Wu et al. [35] fabricated GO-modified self-healing MCs for cardanol-based epoxy anti-corrosion coatings. In their work, it was proposed that The GO's distinct hexagonal ring structure gives it an extremely high barrier property, making it a dense 2D material. The microcapsules' surface will become coated in GO, which will prevent the corrosive medium from passing through them and enhance barrier properties and enhancing corrosion resistance. In this case, the GNP is also a dense structure that would augment the anti-corrosion properties of the coating, allowing for a more robust barrier to corrosive agents.
Comment 22:
Conclusions could be more concise.
Author response 22:
The authors appreciate this comment and have restructured the conclusion to be shorter and more concise, including the relevant information.
Round 2
Reviewer 1 Report
Comments and Suggestions for Authors
The authors have significantly improved their manuscript but I feel that some of the answers to the comments require a bit more modification of the manuscript so that there is complete clarity to what is being claimed.
In particular, the good dispersion of the GNPs in the oil needs to be demonstrated. XPS alone does not show this. If the authors are to claim through their schematic that the GNPs are single particles when reaching the interface, then this must be demonstrated.
The explanation of the usefulness of the contact angle measurements need to be made perfectly clear in the manuscript - how exactly does this relate to the behaviour of the microcapsules on substrates when the droplets used for the measurements have a different structure than the capsules? In their answer to this comment, the authors also state ' This correlation is vital for tailoring the properties of the microcapsules for specific applications.' - it would be good for them to explain (in the manuscript) how they are planning to use these data to tailor the microcapsules properties.
Answer to comment 11 is confusing. The gelatin is likely not at the interface because the surfactant will preferably adsorb because it is more interfacially active and also will have faster kinetics of adsorption. The gelatin will affect the water phase viscosity. In addition, saying that the 'the Gelatin reduces the interfacial dilatational viscosity of the oil phase' is not appropriate. The interfacial dilatational viscosity is a property of the interface and not of one of the phases.. All of these issues need to be addressed in the manuscript.
Comments on the Quality of English Language
It is mostly fine
Reviewer 3 Report
Comments and Suggestions for Authors
After revising the new version of the manuscript, comments were partially covered, and answers were not precise and confusing. Improvements were made in the manuscript, but clarity must be improved. Thus, the manuscript can be considered for publication after major revision and after addressing the following issues:
Comment 6:
The XRD characterization needs to be more precise. Why was the sample prepared at different conditions (see page 3, lines 115-118) compared to the description for the MC formulation (page 2, lines 87-89)? What was the purpose of understanding the interaction between GNP and TO?
Author response 6:
The authors appreciate this comment however the XRD and the microencapsulation process are two different operations. The microencapsulation process was dried at 25 °C in ambient conditions to put them in the epoxy coating, while for the XRD they were dried in the oven to prevent any traces of water in the equipment. The purpose of the XRD was to see any changes in crystallinity in the TO/GNP mixture compared to GNP.
New Comment:
The reviewer understood that the process was different. The question was, why was the sample prepared with a different process? The changes in crystallinity in the TO/GNP compared to GNP are pretty obvious. So, again, what was the purpose of understanding the interaction between GNP and TO? What was expected? How does this influence the purpose of the paper? Another question arises considering that the sample for XRD was done at 80ºC. Is there an effect of temperature? It is assumed that after the process for XRD, the sample was not a powder; thus, was some milling done?
Comment 11:
The critical part is the assumption on page 6, lines 231-232, which reads, “Additionally, the GNP-TO conveyed a homogeneous layer…” Looking at the XRD diffractogram, the width of the peaks is practically the same, just a change in the intensity.
Author response 11:
We agree with the reviewer’s observation about the XRD peaks. This implies the well distributed GNP in the TO medium, and the crystallinity of the GNP is preserved after the mixing.
New comment:
The reviewer disagrees because how does the crystallinity remain if the GNP is well distributed in the TO? Furthermore, why is the diffractogram practically the same? Please explain carefully in coherence with your results and all the suggestions presented in the manuscript, for instance, by microscopy.
Comment 12 is not entirely resolved.
Comment 19:
Looking at the TGA thermogram (Figure 15), results do not make sense, particularly the GNP curve; if this is the TGA thermogram, explain why the XPS, XRD, and Raman results for pure GNP?
Author response 19:
The authors agree with this comment and have subsequently removed the TGA results from the manuscript in terms of further in-depth future studies.
New Comment: The question is still open. The point was not to remove it; the point was to have an explanation of the results. So, please comment.
Comment 21:
There are no proposed mechanisms that explain the obtained results.
Author response 21:
The authors appreciate the comment and also agree the analysis can be improved. The following additions were made to the paper:
Line 336-341: In some similar work [34], the mechanical properties of PUF MCs incorporated with nano-SiO2 were investigated both experimentally and by molecular dynamics simulation. It was also seen here that the SiO2 had a significant effect in augmenting the mechanical properties, similar to the GNP in this case. In their work, the density of the MC increased, and the fractional free volume reduced, which in turn greatly enhanced the mechanical properties of the MCs. A similar mechanism is proposed to also happen in this case, leading to higher apparent toughness of the samples.
Line 374-380: In a similar case, Wu et al. [35] fabricated GO-modified self-healing MCs for cardanol-based epoxy anti-corrosion coatings. In their work, it was proposed that The GO's distinct hexagonal ring structure gives it an extremely high barrier property, making it a dense 2D material. The microcapsules' surface will become coated in GO, which will prevent the corrosive medium from passing through them and enhance barrier properties and enhancing corrosion resistance. In this case, the GNP is also a dense structure that would augment the anti-corrosion properties of the coating, allowing for a more robust barrier to corrosive agents.
New Comment:
The answers and changes in the manuscript do not correspond to a mechanism that explains the results and behavior of the samples. They are rather comparative cases. Please elaborate on an appropriate answer and make the changes in the manuscript.
Comments on the Quality of English LanguageThere are minor details that should be polished
